# Identification of a Rare Novel KMT2C Mutation That Presents with Schizophrenia in a Multiplex Family

**DOI:** 10.3390/jpm11121254

**Published:** 2021-11-25

**Authors:** Chia-Hsiang Chen, Ailing Huang, Yu-Shu Huang, Ting-Hsuan Fang

**Affiliations:** 1Department of Psychiatry, Chang Gung Memorial Hospital-Linkou, Taoyuan 333, Taiwan; yushuhuang1212@gmail.com; 2Department and Institute of Biomedical Sciences, College of Medicine, Chang Gung University, Taoyuan 333, Taiwan; genie.cgu@gmail.com; 3Department of Psychiatry, Yuli Branch, Taipei Veterans General Hospital, Hualien 981, Taiwan; ailing.u8130@msa.hinet.net

**Keywords:** schizophrenia, whole-genome sequencing, rare mutation, KMT2C, histone 3 lysine 4, methylation

## Abstract

Schizophrenia is a complex genetic disorder involving many common variants with modest effects and rare mutations with high penetrance. Rare mutations associated with schizophrenia are highly heterogeneous and private for affected individuals and families. Identifying such mutations can help establish the molecular diagnosis, elucidate the pathogenesis, and provide helpful genetic counseling for affected patients and families. We performed a whole-exome sequencing analysis to search for rare pathogenic mutations co-segregating with schizophrenia transmitted in a dominant inheritance in a two-generation multiplex family. We identified a rare missense mutation H1574R (Histidine1574Arginine, rs199796552) of KMT2C (lysine methyltransferase 2C) co-segregating with affected members in this family. The mutation is a novel deleterious mutation of KMT2C, not reported before in the literature. The KMT2C encodes a histone 3 lysine 4 (H3K4)-specific methyltransferase and involves epigenetic regulation of brain gene expression. Mutations of KMT2C have been found in neurodevelopmental disorders, such as Kleefstra syndrome, intellectual disability, and autism spectrum disorders. Our finding suggests that schizophrenia might be one of the clinical phenotype spectra of KMT2C mutations, and KMT2C might be a novel risk gene for schizophrenia. Nevertheless, the co-segregation of this mutation with schizophrenia in this family might also be due to chance; functional assays of this mutation are needed to address this issue.

## 1. Introduction

Schizophrenia is a complex genetic disorder with the involvement of genetic and environmental factors. Genetic factors play a significant role in the pathogenesis of schizophrenia. Genetic studies of schizophrenia reveal that the genetic architectures of schizophrenia are very diverse, including many common single nucleotides polymorphisms (SNPs) with modest effects, various rare mutations with high clinical penetrance in multiple genes, and aberrant epigenetic regulation of brain gene expression [1,2,3]. Identifying genetic variations associated with schizophrenia can help establish the molecular diagnosis, elucidate the pathogenesis, and provide helpful genetic counseling for affected patients and families. 

In the past, it was a challenging task to identify rare mutations associated with schizophrenia because of their low allele frequencies and high genetic heterogeneity. The advances of genome-wide mutation scanning technologies such as chromosomal microarray [4] and next-generation sequencing [5,6] have facilitated the discovery of rare mutations associated with schizophrenia at the genome-wide level. Hence, increasing numbers of rare mutations with high penetrance were identified in patients with schizophrenia, including aberrant chromosomal numbers and structures [7], copy number variations of genomic DNA segments (CNVs), small insertions/deletions (Indels), and single nucleotide variants (SNVs) [8,9]. Rare mutations associated with schizophrenia are highly heterogeneous and personalized; each affected individual and family have specific mutations [9,10]. Hence, researchers suggested that combining high-throughput sequencing technology and family analysis of large multiplex pedigrees is an effective strategy for identifying novel private mutations associated with schizophrenia and other psychiatric disorders [11,12]. In our series of genetic studies of schizophrenia, we searched for mutations in multiplex families with schizophrenia using systemic genetic approaches, including karyotyping analysis, chromosomal microarray analysis, and next-generation sequencing analysis. Here, we report identifying a rare missense mutation in the KMT2C gene co-segregated with schizophrenia in a multiplex family with the dominant inherited of schizophrenia using whole-exome sequencing.

## 2. Materials and Methods

### 2.1. Subjects

We recruited simplex and multiplex families with schizophrenia into our series of genetic studies of schizophrenia. The study was approved by the Review Board of Chang Gung Memorial Hospital-Linkou with the approval number 201801385A3. Clinical data were collected from interviews with the subjects and reviews of medical records. The psychiatric diagnosis was based on the diagnostic criteria of the Diagnostic and Statistical Manual of Mental Disorder, 5th edition (DSM-5). Informed consent was obtained from each subject after the full explanation of this study. Blood was collected from each participant for genetic experiments. 

### 2.2. Systemic Genome-Wide Mutations Scanning

We performed karyotyping analysis to detect chromosomal abnormalities using the conventional G-banding method. Additionally, we used the oligonucleotide-based whole-genome 385 K CGH arrays (NimbleGen Systems Inc. Madison, WI) to detect pathogenic CNVs. Finally, we conducted whole-exome sequencing using the Illumina HighSeq2000 platform (Illumina, San Diego, CA, USA) or whole-genome sequencing using the Illumina NovaSeq6000 platform (Illumina, San Diego, CA, USA) to detect pathogenic small indels and SNVs. The experiments followed the standard protocols provided by the manufacturer. After a quality check, the raw data of whole-exome sequencing or whole-genome sequencing were aligned to the human reference genome build hg19/GRch37. SAMtools and the Genome Analysis Tool Kit were used to refine the local alignment and generate a variant calling file (VCF) for each subject. Variants were further annotated, filtered, and analyzed under different inheritance modes. The bioinformatics and family analyses were implemented using SeqsLab software (Atgenomics, Taipei, Taiwan, https://doi.org/10.1101/239962, accessed on 22 October 2021). 

### 2.3. Sanger Sequencing

We also used Sanger sequencing to verify the authenticity of the mutations identified from the whole-exome sequencing analysis. In brief, 30 cycles of PCR were performed in a 20 μL mixture containing 100 ng DNA, 1 μM each of the forward primer and the reverse primer, 1X buffer, 0.25 mM of dNTP, and 0.5 U of Power Taq polymerase (Genomics, New Taipei City, Taiwan). An aliquot of the amplicon was purified and subjected to Sanger sequencing using the BigDye Terminator kit v3.1 (Applied Biosystems, Foster, CA, USA).

### 2.4. Bioinformatics Analysis

The frequency of the mutation identified in this study was checked in the dbSNP (https://www.ncbi.nlm.nih.gov/snp/, accessed on 22 October 2021) and the Taiwan biobank (https://taiwanview.twbiobank.org.tw/index, accessed on 22 October 2021). To assess the functional impact of this mutation, we conducted in silico analysis using several online software, including Polyphen-2 (http://genetics.bwh.harvard.edu/pph2/index.shtml, accessed on 22 October 2021), SIFT (https://sift.bii.a-star.edu.sg/, accessed on 22 October 2021), and PROVEAN (http://provean.jcvi.org/index.php, accessed on 22 October 2021).

## 3. Results

### 3.1. Recruitment of a Schizophrenia Multiplex Family

We recruited a two-generation family with five affected patients diagnosed with schizophrenia into this study. As shown in Figure 1, the inheritance of schizophrenia in this family matches the dominant pattern. After giving birth to her first daughter, the affected mother (I-2) had a mental illness in her twenties. She manifested symptoms of self-talking, bizarre behavior, and delusion of persecution and was diagnosed with schizophrenia. She gave birth to five children in total. Except for the fourth daughter (II-4), who was free of mental illness, all four other children were diagnosed with schizophrenia in their twenties. They had similar psychotic symptoms to their mother. Before their mental illnesses, they all had normal psychosocial development and did not have other physical and behavioral problems. After the onset of their mental illnesses, their social and working functions deteriorated gradually. The socio-demographic data and disease-related information are summarized in Table 1.

### 3.2. Identification of Rare, Likely Pathogenic Mutations

The conventional karyotyping analysis did not find chromosomal abnormalities in this family. Additionally, the oligonucleotide-based whole-genome microarray analysis did not find pathogenic CNVs completely segregated with schizophrenia in this family (Figure 1). In the whole-exome sequencing analysis, we first compared the data of three affected members (II-2, II-3, and II-5) with five unrelated controls. We obtained 85 variants present only in the three patients. Then, we filtered the patient-specific variants using the criteria of allele frequency less than 0.001, the likely pathogenic predicted by the Atgenomics software, and genetic function related to neurobiology and brain. We obtained four mutations meeting the criteria. The genetic information of these four mutations is summarized in Table 2. 

### 3.3. Sanger Sequencing and Segregation Analysis

To verify the authenticity of these four rare, likely pathogenic mutations and conduct a family analysis, we designed primer pairs covering each mutation and performed PCR-based Sanger sequencing. The sequences of the primer pairs, optimal annealing temperature, and amplicon sizes are listed in Table 3.

In the family analysis, we found that the mutations of SHANK2, CSMD1, and RPH3A did not segregate completely with affected members in this family. We found that only the H1574R (Histidine1574Arginine) mutation of KMT2C co-segregated completely with affected members in this family. The segregation analysis results are shown in Figure 2, and the Sanger sequencing results of the wild-type and the H1574R mutation of KMT2C are shown in Figure 3.

### 3.4. Bioinformatics Analysis

The H1574R mutation of KMT2C was assigned rs199796552 in the dbSNP. The frequency of this allele was very rare in several public genome databases, including 0.002307 in the Taiwan biobank, 0.000189 in the Exome Aggregation Consortium (ExAC), 0.000043 in the Genome Aggregation Database (GnomAD), and 0.000096 in the Allele Frequency Aggregator (ALFA), respectively. The mutation was predicted to be deleterious by PROVEAN (score −3.51), tolerated by SIFT (score 0.271), and probably damaging by Polyphen-2 (score 0.999), respectively. 

## 4. Discussion 

The KMT2C gene encodes a methyltransferase, a myeloid/lymphoid or mixed-lineage leukemia (MLL) family member. The MLL family consists of KMT2A, 2B, 2C, 2D, 2F, and 2G genes. These six genes catalyze histone 3 lysine 4 (H3K4) methylation and epigenetically regulate many gene expressions in the brain through chromatin modification [13]. Dysregulation of H3K4 methylation in the brain has been implicated in several neurodevelopmental disorders, including autism and schizophrenia [14,15]. Rare KMT2C mutations have been found in patients with intellectual disability [16,17], autism spectrum disorders [18,19,20], schizophrenia [21], non-syndromic primary failure of tooth eruption [22], and a family with colorectal cancer and acute myeloid leukemia [23], indicating the pleiotropic effects of KMT2C mutations. 

The H1574R mutation of KMT2C reported in this study is a novel and private mutation for this multiplex schizophrenia family, and it was not reported in the literature to our knowledge. Senormanci and colleagues conducted whole-exome sequencing of several candidate genes in two families with schizophrenia. They detected damaging indels in several genes, including SPON1 (spondin 1), GRIA3 (glutamate ionotropic receptor AMPA tpe subunit 3), SMAD5 (SMAD family member 5), KMT2C, SRD5A2 (steroid 5-alpha-reductase 2), SEMA3B (semaphoring 3B), NCOR2 (nuclear receptor corepressor 2), GPHB5 (glycoprotein hormone subunit beta 5), FAM174B (family with sequence similarity 174 member B), CLTCL1 (clathrin heavy chain like 1), and TMEM216 (transmembrane protein 216) [21]. The co-segregation of the H1574R mutation of KMT2C in this multiplex family suggests that the KMT2C gene might be a novel risk gene for schizophrenia. Nevertheless, it is also likely due to chance only, and hence, it is necessary to perform experiments to investigate the pathogenicity of this mutation in the future study.

KMT2F (also known as SETD1A (SET domain containing 1A, histone lysine methyltransferase)) is a member of the MLL gene family like KMT2C. Several studies have identified rare KMT2F mutations in patients with schizophrenia [24,25,26]. Guipponi and colleagues detected a de novo splice mutation of KMT2F from exome sequencing analyses of 53 trio families with sporadic cases of schizophrenia [24]. Takata and colleagues conducted exome sequencing analyses of 231 trio families with schizophrenia and 34 control trios; they detected two de novo loss-of-function mutations of KMT2F. Hence, they suggested that the KMT2F was a candidate susceptibility gene of schizophrenia [26]. Singh and colleagues analyzed whole-exome sequencing data consisting of 4264 schizophrenia cases, 9343 controls, and 1077 trios. They identified 10 rare loss-of-function KMT2F mutations in patients with schizophrenia, which reached a genome-wide significant association compared with controls [25]. They further detected rare KMT2F mutations in children with severe developmental disorders and other neuropsychiatric phenotypes [25]. Hence, the research group concluded that loss-of-function KMT2F mutations caused several neurodevelopmental disorders, including schizophrenia [25]. They also suggested that epigenetic dysregulation, especially in the methylation of the H3K4 pathway, was an essential mechanism of the pathogenesis of schizophrenia [25]. 

The KMT2C is also a member of the MLL family specifically involved in the H3K4 methylation. If functional assays of the H1574R mutation of KMT2C show pathogenic effects in future studies, our findings would be in line with previous studies that detected KMT2F mutations in schizophrenia and support the idea that dysregulation of H3K4 is a molecular mechanism of schizophrenia. 

## Figures and Tables

**Figure 1 jpm-11-01254-f001:**
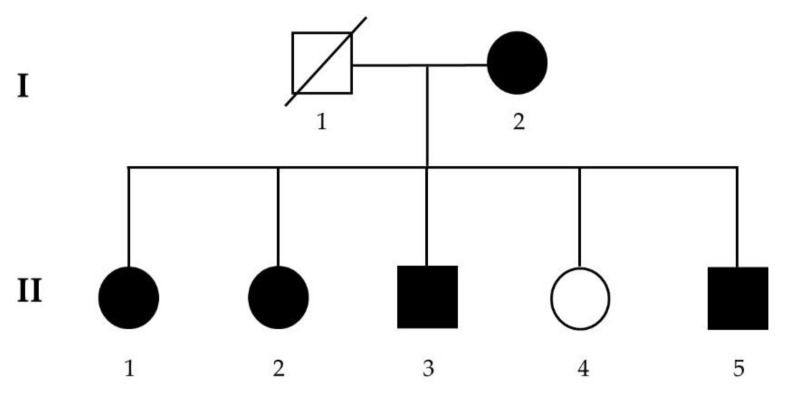
Pedigree of the two-generation multiplex family with schizophrenia.

**Figure 2 jpm-11-01254-f002:**
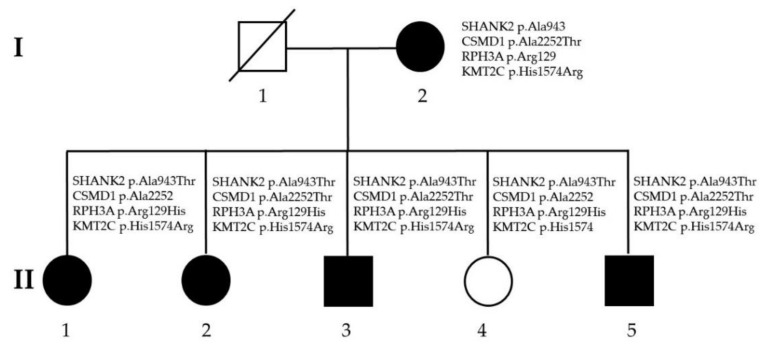
Family analysis of four rare, likely pathogenic mutations identified in this study.

**Figure 3 jpm-11-01254-f003:**
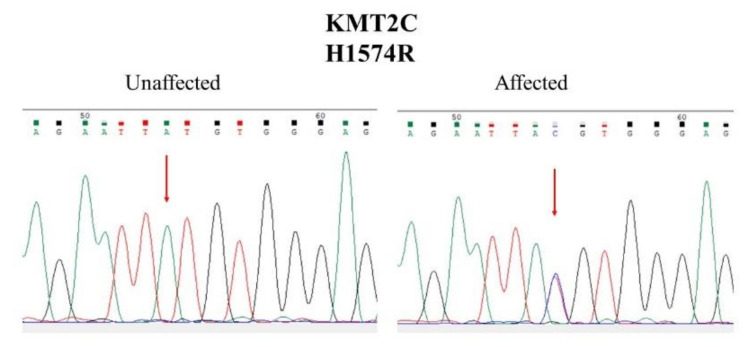
Representative results of Sanger sequencing of the H1574R mutation of KMT2C in affected members and the wild-type mutation of KMT2C in the unaffected member.

**Table 1 jpm-11-01254-t001:** Summary of socio-demographic data and disease-related data of the family in this study.

	I-2	II-1	II-2	II-3	II-4	II-5
Age (years)	66	47	44	43	41	39
Gender	Female	Female	Female	Male	Female	Male
Education	Primary school	Primary school	Junior high school	Junior high school	Junior high school	Junior high school
Marital status	Married	Unmarried	Unmarried	Unmarried	Married	Unmarried
Diagnosis	Schizophrenia	Schizophrenia	Schizophrenia	Schizophrenia	Normal	Schizophrenia
Age of onset (years)	20	20	24	22		22
Main psychotic symptoms	Self-talking, auditory hallucination, persecutory delusion	Self-talking, auditory hallucination, unstable mood	The idea of reference, persecutory delusion, irrelevant speech	Auditory hallucination, religious and persecutory delusions		Auditory hallucination, persecutory delusion
Social function	Poor, staying in a chronic mental hospital	Poor, staying in a chronic mental hospital	Poor, staying in a chronic mental hospital	Poor, staying in a chronic mental hospital	Good, holding a job	Poor, staying in a chronic mental hospital
Co-morbidity	No	No	No	No	No	No

**Table 2 jpm-11-01254-t002:** Rare, likely pathogenic mutations identified in this family.

Gene	Chromosome	Mutation	dbSNP	Allele Frequency
SHANK2	11q13.3-q13.	NC_000011.9:g.70333379C>T NM_012309.4:c.2827G>A NM_012309.4:p.Ala943Thr (Alanine943Threonine)	rs781910453	0 in ALPHA and Taiwan biobank
CSMD1	8p23.2	NC_000008.10:g.2966125C>T NM_033225.5:c.6754G>A NM_033225.5:p.Ala2252Thr (Alanine2252Threonine)	rs369177851	0.000087 in ALPHA and 0 in Taiwan biobank
RPH3A	12q24.13	NC_000012.11:g.113304599G>A NM_014954.3:c.386G>A NM_014954.3:p.Arg129His (Arginine129Histidine)	rs148899308	0 in ALPHA and 0.001648 in Taiwan biobank
KMT2C	7q36.1	NC_000007.13:g.151884872T>C NM_170606.2:c.4721A>G NM_170606.2:p.His1574Arg (Histidine1574Arginine)	rs199796552	0.000096 in ALPHA and 0.002 in Taiwan biobank

SHANK2: SH3 and multiple ankyrin repeat domains 2; CSMD1: CUB and Sushi multiple domains 1; RPH3A: rabphilin 3A; KMT2C: Lysine methyltransferase 2C; ALFA: Allele Frequency Aggregator.

**Table 3 jpm-11-01254-t003:** Primer sequences, optimal annealing temperature (Ta), and sizes of amplicons of Sanger sequencing in this study.

Mutation	Primer (5‘-3‘)	Ta (℃)	Amplicon Size (bp)
SHANK2 p.Ala943Thr	Forward: GCTGCTCTTGCCGCTGCTGCTGG Reverse: TACAGTCGGAATGCCGGCCCGCA	60	246
CSMD1 p.Ala2252Thr	Forward: ACAGGAAACGCTGGCCTCCAGTG Reverse: TGGCAACACAGCCCTCGAAACGG	60	253
RPH3A p.Arg129His	Forward: GGCTACGTTGCCATGCCTCCCAT Reverse: ACCTCTTAAGGCACCTCTTAAGC	60	224
KMT2C p.His1574Arg	Forward: AAGGCTTGTACCTGGCATCAGGA Reverse: TGTGTTTATGTGGACCCATGCAG	60	257

## Data Availability

The data presented in this study are available on request from the corresponding author. The data are not publicly available due to privacy.

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
