# Peer review of "Identification of a Rare Novel KMT2C Mutation That Presents with Schizophrenia in a Multiplex Family"

_jpm, 2021, doi:10.3390/jpm11121254_

Round 1

Reviewer 1 Report

The authors of this manuscript report the rare KMT2C gene mutation they found in 1 family in which several patients with schizophrenia. I would like more information on the course of the disease in different members of this family. For example,  age of onset, severe, and what symptoms prevailed. Very little socio-demographic data regarding the patients and their relatives studied.  Did the subjects have any other chronic diseases? Did they have any bad habits, body mass index, etc. Nothing about this is mentioned. Nor does it say anything about their social status. How many people in total were in the study? It seems better to present all this information in a table form.

The finding of a mutation associated with schizophrenia in a study of a family with several patients is very interesting. But it begs the question. Were there other mutations associated with schizophrenia in this particular family?

I would like clarification about the statistical programs that were used to calculate the sequencing results.

On the whole, it is an interesting work.

Author Response

  1. The family has six members, including the mother and her five children. Except for one daughter who was free of psychiatric diagnosis, all the other five members were diagnosed with schizophrenia. We were able to study all the members of this family. We summarize the socio-demographic and disease-related data in the new Table 1 as suggested.
  2. We indeed found the other pathogenic mutations implicated in psychiatric disorder in his family, but they did not co-segregate with schizophrenia in this family. We summarized these mutations in the new Table 2 as suggested.
  3. For next-generation sequencing analysis, we used the pipeline SeqsLab: an integrated platform for cohort-based annotation and interpretation of genetic variants on Spark, which is established by the Atgenmics (Taipei, Taiwan) (https://doi.org/10.1101/239962). We also add this information in our revised method section 2. Systemic genome-wide mutations scanning.
  1. We also add more information about mutation scanning and segregation analysis in the revised Results section.i

Reviewer 2 Report

The authors attempted to identify rare mutations associated with schizophrenia incidence. They used multiple methods to find rare mutations co-segregating with schizophrenia patients in a two-generation multiplex family with seven members. The authors find a single rare missense mutation of the KMT2C gene co-segregating with affected family members with whole-exome sequencing.

Generally, if this article provides more data supporting their novel rare variant on KMT2C is associated with schizophrenia, it would be more helpful to convince readers.

In section 2, the authors mentioned that they recruited simplex and multiplex families for the research and performed multiple genome-wide variant scanning experiments. However, the statistics of the recruited family nor the descriptions of variant scanning results are missing.

For example, it would be more convincing if the article provides data such as the number of variants identified from each individual, the number of co-segregated variants among schizophrenia patients, the number of pathogenic co-segregated variants.

In addition, the authors claim that the H1574R mutation of KMT2C is a significant risk mutation for schizophrenia. To support this, the authors can show whether the affected family members do not have the other genetic variants associated with schizophrenia. Moreover, if the co-segregated variants among affected family members other than the KMT2C variant do not contain different pathogenic variants, the novel KMT2C variant's association to schizophrenia would be more convincing.

The authors also indicated that the KMT2C involves epigenetic regulation of brain gene expression. If the authors provide gene expression change profiles according to the H1574R mutation of KMT2C in neuronal cell-line, this can be more confirmatory data for the effect of the rare KMT2C variants.

I suggest the authors prepare more figures and numbers to support their novel KMT2C variant's association with schizophrenia. 

Author Response

  1. The current paper is a part of our ongoing genetic study of schizophrenia. The goal of this study is to find personalized genetic deficits for each family based on the rare mutation hypothesis of schizophrenia. Because research is still ongoing; hence, we did not provide the total number of families in this report.
  2. In our study, we indeed find the other rare, likely pathogenic mutations implicated in psychiatric disorders in this family. But they did not co-segregate with the affected members in this family. Hence, we summarized these mutations in the new Table 2 in the revised manuscript. Also, we add a new Figure 2 to show the segregation analysis of rare, likely pathogenic mutation in the revised manuscript.
  3. Due to the limited resources and expertise of the authors, we are not able to carry out the expression as suggested at this moment. We mention this limitation in the revised manuscript as “Identifying the H1574R mutation of KMT2C co-segregating with affected members in this multiplex family indicates that the KMT2C gene is likely a risk gene for schizophrenia. It is necessary to carry out functional assays of this mutation in a future study to test its pathogenicity.”

Round 2

Reviewer 2 Report

The authors revised the manuscript to provide more evidence of their novel schizophrenia variant regarding my suggestions. 

However, the authors did not present the effect of their KMT2C H1574R variant on the gene expression in neuronal cells. It is essential to provide more experiment results because other risk factors of schizophrenia, such as environmental effects, were not controlled; with the materials in the manuscript, we can't conclude the variant is associated with schizophrenia. Thus, the authors need to perform experiments to show the expression change due to their novel variant. It is because the novel variant can be found by chance and may have no effects on schizophrenia.

If the authors can not present the experiment result in the meantime, I suggest revising the manuscript to more cautiously mention the relation between the novel variant and the schizophrenia incidence.

Author Response

Thanks for the valuable comment. We cannot perform functional assays to verify the pathogenicity of this novel mutation at this stage. So, we follow the suggestion and revise our manuscript to be more cautiously to mention the relationship between this novel variant and the schizophrenia incidence.

In the Abstract, we revise the last sentence to "Our finding suggests that schizophrenia might be one of the clinical phenotype spectra of KMT2C mutations, and KMT2C might be a novel risk gene for schizophrenia. Nevertheless, the co-segregation of this mutation with schizophrenia in this family might also be due to chance; functional assays of this mutation are needed to address this issue." (Line 24-27).

In the Discussion, we revise the last sentence of the second paragraph to "The co-segregation of the H1574R mutation of KMT2C in this multiplex family suggests that the KMT2C gene might be a novel risk gene for schizophrenia. Nevertheless, it is also likely due to chance only, and hence, it is necessary to perform experiments to investigate the pathogenicity of this mutation in the future study." (line 177-181).

We revised the second sentence to "The KMT2C is also a member of the MLL family specifically involved in the H3K4 methylation. If functional assays of the H1574R mutation of KMT2C show pathogenic effects in future studies, our findings would be in line with previous studies that detected KMT2F mutations in schizophrenia and support the idea that dysregulation of H3K4 is a molecular mechanism of schizophrenia." (line 199-203)

Round 3

Reviewer 2 Report

Through a couple of revisions, the authors made the manuscript more explicit. As suggested in the previous reviews, this article can provide more enlightening results for schizophrenia research and other neurodegenerative disease research if this manuscript accompanies by more functional study results. However, I think it is worth enough sharing with the schizophrenia research community that the authors identified a possible pathogenic risk variant co-segregated with schizophrenia patients within a two-generation multiplex family. Based on this novel possible pathogenic risk variant, I expect more comprehensive and detailed results regarding the schizophrenia risk factors from future studies.